# The Adaptive Immune Response in Hepatitis B Virus-Associated Hepatocellular Carcinoma Is Characterized by Dysfunctional and Exhausted HBV-Specific T Cells

**DOI:** 10.3390/v16050707

**Published:** 2024-04-29

**Authors:** Malene Broholm, Anne-Sofie Mathiasen, Ása Didriksen Apol, Nina Weis

**Affiliations:** 1Department of Infectious Disease, Copenhagen University Hospital, 2650 Hvidovre, Denmark; 2Department of Clinical Medicine, Faculty of Health and Medical Sciences, University of Copenhagen, 2300 Copenhagen, Denmark

**Keywords:** hepatitis B virus, hepatocellular carcinoma, immune response, HBV-specific T-cell response

## Abstract

This systematic review investigates the immunosuppressive environment in HBV-associated hepatocellular carcinoma (HCC), characterized by dysfunctional and exhausted HBV-specific T cells alongside an increased infiltration of HBV-specific CD4+ T cells, particularly regulatory T cells (Tregs). Heightened expression of checkpoint inhibitors, notably PD-1, is linked with disease progression and recurrence, indicating its potential as both a prognostic indicator and a target for immunotherapy. Nevertheless, using PD-1 inhibitors has shown limited effectiveness. In a future perspective, understanding the intricate interplay between innate and adaptive immune responses holds promise for pinpointing predictive biomarkers and crafting novel treatment approaches for HBV-associated HCC.

## 1. Introduction

Hepatocellular carcinoma (HCC) is a leading cause of cancer-related deaths globally [1,2], with 55–80% of the cases linked to chronic hepatitis B virus (CHB) [3], which affects approximately 296 million individuals worldwide [4]. The natural history of CHB varies widely among patients, with some entering a phase of low viral replication and limited inflammation, while others may have persistent inflammation leading to tumorigenesis. Although the precise mechanisms determining disease outcomes remain unclear, there is growing evidence indicating that the prognosis of CHB relies on the interplay between the virus and the host immune response. The adaptive immune response is crucial for viral clearance, but the destruction of virus-infected cells also induces inflammation and liver damage.

During viral infections, CD4+ T cells primarily differentiate into T helper (Th1) effector cells, which induce the activation of cytotoxic CD8+ T cells, and into T follicular (Tfh) helper cells which activate B cells [5,6]. Additionally, CD4+ T cells can differentiate into regulatory T cells (Tregs), a specialized population that suppresses the activation, proliferation, and effector functions of multiple immune cells, including T cells, B cells, natural killer cells and dendritic cells [7]. This regulatory function is essential for maintaining self-tolerance. However, an expanded population of Tregs can inhibit the antiviral immune response, leading to persistent inflammation, and even suppress anti-cancer immunity and protective immune surveillance [8].

During acute HBV infection, the HBV-specific CD8+ T-cell response is usually strong and polyclonal, characterized by the production of pro-inflammatory cytokines such as INF-γ and TNF-α, and cytotoxic molecules such as granzyme and perforin, which are essential for controlling HBV infection [9]. Although levels of HBV-specific CD8+ T cells usually decline and become undetectable during the acute phase [10], these cells persistently remain detectable in CHB, albeit usually at low levels [9]. Previous research has suggested that the persistence of HBV-specific CD8+ T cells may contribute to HBV-induced carcinogenesis by maintaining inflammation and accelerating hepatocyte turnover, thereby promoting fibrosis and cirrhosis [10,11,12,13,14].

Moreover, in CHB, the adaptive immune response is characterized by a dysfunctional and exhausted T-cell response marked by the expression of multiple inhibitory receptors, including programmed cell death-1 (PD-1), cytotoxic T-lymphocyte antigen 4 (CTLA-4), and T-cell immunoreceptor with Ig and ITIM domains (TIGIT). This induces an immunosuppressive microenvironment leading to persistent inflammation and ultimately contributes to tumorigenesis [8,15,16,17,18]. Characterization of the immune response and complex interactions within the tumor microenvironment is essential for understanding the mechanisms of disease progression and for the future development of new targeted therapies. This review aims to explore the role of the adaptive immune response in HBV-associated HCC and to discuss the mechanisms involved in the immunosuppressive tumor microenvironment.

## 2. Materials and Methods

### 2.1. Search Strategy

This systematic review was conducted and reported according to the Preferred Reporting Items for Systematic Reviews (PRISMA) 2020 statement [19]. The PRISMA 2020 replaces the PRISMA 2009 guideline with an expanded 27-item checklist. The study was registered at the international prospective register of systematic reviews (PROSPERO) as recommended by the Cochrane Collaboration (registration ID: CRD42023493591). A search strategy was developed and the PubMed/Medline, Cochrane Library, and Embase databases were systematically searched on 1 December 2023 for original studies investigating T-cell activity in patients with HBV-related HCC. We used the following search string: hepatitis B virus OR HBV AND hepatocellular carcinoma OR carcinoma OR cancer OR tumor OR HCC AND T cell OR CD4 OR CD8 OR Treg OR regulatory T cell.

We also manually searched reference lists of relevant articles for additional studies. Assessment of abstracts and inclusion of studies was performed independently by two reviewers (M.B. and A.S.M). Any differences were settled through discussion.

### 2.2. Eligibility Criteria

All study types were eligible for inclusion except for reviews and meta-analyses. It was not obligatory for the studies to include control groups.

Inclusion criteria were as follows:Age of study population ≥ 18 years;Studies in English language;Only studies including humans;Studies including patients with HBV-associated HCC;

Exclusion criteria were as follows:Pediatric patients < 18 years of age;Studies only performed on animals;Reviews or meta-analyses;Studies on individuals with HBV infection without a group of individuals with HBV-related HCC;Studies including CHB patients coinfected with other hepatitis viruses and/or human immunodeficiency virus (HIV).

### 2.3. Comparison Groups

We divided comparison groups into three groups: (1) a group of CHB patients, which were defined as chronic HBV infection with or without liver cirrhosis, but without HCC; (2) a group of patients with non-HBV HCC involving patients with HCC with HCV infection and/or non-viral etiologies; (3) the healthy comparison group was defined as patients without hepatitis infection, HCC, alcohol abuse, or other liver diseases.

## 3. Results

Our search strategy resulted in 1411 studies of which 52 were included in this systematic review, collectively involving a total of 6931 participants. The study participants were categorized into four groups of patients as follows: (1) HBV-associated HCC (*n* = 3660), (2) CHB without HCC (*n* = 740), (3) non-HBV HCC (*n* = 1627), and (4) healthy comparison (*n* = 650). Among the included studies, 47 conducted a comparative analysis. Studies were clinical translational studies, with 36 studies focusing on local biomarkers in formalin-fixed paraffin-embedded (FFPE) or fresh–frozen tissue samples, and 38 studies investigating systemic biomarkers in peripheral blood mononuclear cells (PBMCs). Furthermore, 21 studies conducted combined analysis of local and systemic biomarkers (Table 1). The study selection process is shown in Figure 1.

### 3.1. Patient Characteristics

The included patients were chronically infected with HBV, defined as persistence of HBsAg in serum > 6 months. In 22 studies, a control group of CHB patients (*n* = 740) was included, 18 studies included a group of patients with non-HBV HCC (*n* = 1627), and 26 studies included a healthy comparison group without HBV infection. The study populations were dominated by male patients, and several studies also included subgroups based on different clinical stages of CHB and/or HCC.

#### 3.1.1. HBV-Specific CD8+ T Cells

A total of 36 studies analyzed HBV-specific CD8+T cells in patients with HBV-associated HCC, of which 24 studies included circulating CD8+ T cells in PBMCs, and 26 studies analyzed infiltrating CD8+ T cells in liver tissue samples.

#### 3.1.2. Phenotypes of Circulating CD8+ T Cells in Patients with HBV-Related HCC

Total levels of global circulating CD8+ T cells were significantly higher in patients with HBV-associated HCC compared with healthy comparisons [41,54], and levels of terminally differentiated effector CD8+ T cells were higher in HBV-associated HCC [21] with the majority being effector CD8+ T cells and memory CD8+ T cells [21,34]. The HBV-specific CD8+ T cells were characterized by exhausted phenotypes with high expression levels of inhibitory markers. In contrast to the comparison group, patients with HBV-associated HCC had markedly higher levels of exhaustion markers, including PD-1 [33,38,61,65], TIGIT [21,38,39], and CTLA-4 expression [39].

In most of the studies, circulating HBV-specific CD8+ T-cell levels were similar between CHB patients and patients with HBV-associated HCC [21,32,41,59,61], whereas one study showed lower levels in HBV-associated HCC [50]. Interestingly, HBV-specific CD8+ T cells showed stronger cytotoxicity in CHB patients [32] and a more exhausted functional phenotype in patients with HBV-associated HCC [67]. Additionally, increased levels of HLA-DR were seen in HBV-associated HCC which indicates highly activated CD8+ T cells [67].

Studies reported significantly higher expression of exhaustion markers, such as PD-1 [38,59,61,65,67], TIGIT [38], and T-cell immunoglobin and mucin domain 3 (TIM-3) [59] in patients with HBV-associated HCC compared with CHB. A study reported that PD-1 expression especially was seen in central memory and effector memory CD8+ T cells [38]. Furthermore, a study found elevated expression of lymphocyte activation gene 3 (LAG-3), CTLA-4, and TIM-3 on PD-1+CD8+ T cells in patients with HBV-associated HCC compared with CHB [59]. Expression of INF-γ decreased in HBV-associated HCC [59] and HBV-specific CD8+ T cells showed impaired capacity for TNF-α secretion [67] in comparison with CHB patients.

Total levels of global circulating CD8+ T cells were elevated in HBV-associated HCC compared with non-HBV HCC [30], and a study reported lower levels of INF-γ, TNF-α, and granzyme B secretion in patients with HBV-associated HCC compared with non-HBV HCC [37], indicating impaired effector function of CD8+ T cells.

#### 3.1.3. Prognostic Value of PD-1 Expression on Circulating CD8+ T Cells

Expression of PD-1 on CD8+ T cells was highly correlated with disease progression [38,61,65] and the association was shown to be stronger in CD8+ T cells than in CD4+ T cells [38]. In addition, high levels of PD-1 were related to higher recurrence rates of HCC [38]. Finally, PD-1 levels were positively correlated with HBV DNA, alanine aminotransferase (ALT), and aspartate aminotransferase (AST) [61].

#### 3.1.4. Tumor-Infiltrating HBV-Specific CD8+ T-Cell Phenotypes in HBV-Associated HCC

The infiltration of CD8+ T cells was markedly lower in the tumor tissue of HBV-associated HCC compared with healthy comparisons or paired non-tumor tissue [40,69,71]. Furthermore, one study reported significantly elevated levels of PD-1 expression in HBV-associated HCC compared with the normal comparison group [65], and a second study reported increased expression of LAG-3 followed by decreasing levels of INF-γ, which reduces the effector function of CD8+ T cells [63]. In addition, a study demonstrated low levels of granzyme A, granzyme B, and perforin in patients with HBV-associated HCC compared with healthy comparisons [69].

The CD8+ T-cell infiltration in HBV-associated HCC was compared within liver tissue from CHB patients in two studies, which reported higher levels of PD-1 on CD8+ T cells in HBV-associated HCC in one study [65], while no difference was found in the second study [29]. Additionally, high expression of TIGIT and CTLA-4 was comparable in the two groups [29].

When comparing tumor tissue from HBV-associated HCC with non-HBV HCC, studies reported higher levels of CD8+ T cells in HBV-associated HCC [47,71]. Tumor-infiltrating CD8+ T cells in HBV-associated HCC were characterized by marked clonal expansion and an immunosuppressive and exhausted tumor microenvironment. Studies demonstrated higher levels of PD-1 [22,24,30,37,47,71], TIGIT [22], CTLA-4 [20,37,71], TIM-3 [30,37], LAG-3 [22], and TOX [22,30] in HBV-associated HCC compared with non-HBV HCC.

#### 3.1.5. Impaired Effector CD8+ T Cell Function in HBV-Associated HCC

The HBV-specific CD8+ T cells are crucial in controlling HBV infection and tumor-infiltrating CD8+ T cells in HBV-associated HCC were positively correlated with survival [30,31,47]. The CD8+ T cells were primarily effector cells and memory cells, and one study reported that effector cells in HBV-associated HCC were characterized by increased levels of CTLA-4, ICOS, and TOX expression [20]. In addition, PD-1 and TIM-3 levels proved to be correlated with poor prognosis [47,65], and PD-1 levels increased with severity of liver fibrosis [27]. Cytotoxic CD8+ T cells in HBV-associated HCC secreted lower levels of INF-γ, TNFα, and granzyme B, compared with non-HBV HCC [37,47]. However, a study reported that inhibition of TIM-3 and PD-1 restored CD8+ T cell function with increasing INF-γ and TNF-α levels, indicating a potential therapeutic target [42]. Finally, a study found decreasing INF-γ levels with increasing LAG-3 expression, suggesting that LAG-3 reduces the effector function of CD8+ T cells [63].

### 3.2. HBV-Specific CD4+ T Cells

A total of 35 studies investigated CD4+ T cells in HBV-associated HCC, of which 25 studies included circulating CD4+ T cells in PBMCs and 24 studies included infiltrating CD4+ T cells in liver tissue samples.

#### 3.2.1. Circulating CD4+ T Cells in HBV-HCC

A study reported decreased levels of total global circulating CD4+ T cells in patients with HBV-associated HCC (*n* = 715) compared with healthy controls (*n* = 100) [54]. In addition, a study showed lower levels of naïve CD4+ T cells compared with a healthy comparison group [21], whereas another study reported increased levels of cytotoxic CD4+ T cells, characterized by granzyme A and B expression [53]. Furthermore, studies found increased levels of PD-1 [33,61] and TIM-3 [21] expression compared with healthy comparisons.

In comparison with CHB patients, total levels of global circulating CD4+ T cells [50] and naïve CD4+ T cells [21] were lower in HBV-associated HCC. The studies found varying results regarding exhaustion markers on circulating CD4+ T cells in patients with CHB compared with HBV-associated HCC. A study found increased expression of TIGIT and TIM-3 in HBV-associated HCC [21]. Expression of PD-1 levels was analyzed in one study and reported similar PD-1 expression between the two groups [61]. It was demonstrated that PD-1 expression on circulating CD4+ T cells was correlated with HBV DNA and ALT levels [61], whereas a second study only found a correlation between PD-1 expression and HBV DNA in CHB patients, but not in patients with HBV-associated HCC [33]. In addition, a study showed elevated HLA-DR in HBV-associated HCC compared with CHB indicating a higher T-cell activation [50].

Only a single study compared total levels of circulating CD4+ T cells in patients with HBV-associated HCC (*n* = 22) and non-HBV HCC (*n* = 17) and found no difference [60]. A second study demonstrated that levels of CD4+ memory T cells were an independent predictor for survival in patients with HBV-associated HCC [23].

#### 3.2.2. Tumor-Infiltrating CD4+ T Cells

Levels of infiltrating CD4 T cells in HBV-associated HCC compared with healthy controls were investigated in two studies finding elevated total CD4+ T cells [68], but decreased cytotoxic CD4+ T cells in HBV-associated HCC [53].

The total levels of infiltrating CD4+ T cells in CHB were not reported across the included studies, and a single study found that total CD4+ T-cell levels were higher in HBV-associated HCC compared with non-HBV HCC [71]. Furthermore, exhausted states of CD4+ T cells were markedly higher in HBV-associated HCC than non-HBV HCC, with increased CTLA-4 and PD-1 expression levels [20,71].

A study including 1328 patients showed that CD4+ T-cell subsets were more enriched in HBV-associated HCC tissue than CD8+ T cells [40]. Among CD4+ T-cell populations, Tregs were the predominant subset and highly enriched in the tumor microenvironment of HBV-associated HCC [25,71], with the highest accessibility at the forkhead box P3 (FOXP3) loci [25].

#### 3.2.3. Circulating Regulatory T Cells in HBV-Associated HCC

Circulating Tregs were primarily characterized as CD4+CD25+ T cells and studies found significantly elevated levels in HBV-associated HCC compared with healthy controls [20,36,41,48,58,66,69,70]. In addition, studies found markedly elevated expression levels of FOXP3 [20,41,66,69,70], PD-1 [21], and TIGIT [21].

Levels of Tregs in HBV-associated HCC compared with CHB showed varying results, as two studies found no difference [41,66] and one study found elevated levels in HBV-associated HCC [69]. In comparison with non-HBV HCC, levels of circulating Tregs were significantly increased in HBV-associated HCC [25,37,60] and expression levels of FOXP3 [25,37], CTLA-4, LAG-3, and PD-1 [37] increased in patients with HBV-associated HCC.

#### 3.2.4. Tumor-Infiltrating Regulatory T Cells in HBV-Associated HCC

Levels of infiltrating Tregs in HBV-associated HCC compared with control tissue were analyzed in three studies and in all cases markedly increased levels of Tregs were demonstrated in HBV-related HCC tumor tissue [36,69,70]. Increased levels of Tregs were also found in HBV-associated HCC in comparison with liver tissue from patients with CHB without HCC [60,69].

In three studies, higher levels of Tregs were documented in HBV-associated HCC compared with HBV-non-associated HCC [26,37,60]. Analyses of Tregs in the HBV-associated HCC tumor microenvironment showed a more immunosuppressive and effective status [20], with increased expression levels of FOXP3 [20,26,37], PD-1 [20,25,37], CTLA-4 [20,25], ICOS [20], and LAG-3 [37] compared with non-HBV HCC.

#### 3.2.5. Role of Regulatory T Cells HBV-Associated HCC

It was demonstrated that levels of circulating Tregs increased with advancing stages of HBV-associated HCC, thus being a predicter for prognosis [69]. A study isolated circulating Tregs from patients with HBV-associated HCC to analyze the function of this cell subset and found that Tregs significantly inhibit proliferation [66]. Additionally, a study showed that Tregs in HBV-associated HCC were more suppressive than in non-HBV HCC, through high IL-10 and TGF-β secretion [60]. Moreover, Tregs suppressed CD8+ T-cell proliferation through the inhibition of INF-γ, TNF-α, and reduced HLA-DR [69]. Another interesting finding was that the CD4+ T and CD8+ T-cell interactions were increasingly replaced with rising Treg level in HBV-associated HCC. Spatial proteomics demonstrated increasing cell–cell connections between CD4+ T cells and Tregs in the tumor microenvironment, and distances between these cells became significantly shorter in the mid tumor regions compared to non-tumor regions [24]. Furthermore, spatial proteomics visualized that PD-1+CD8+ T cells also connected with Tregs [24]. Finally, a study demonstrated that the Treg/CD8+ ratio was strongly associated with prognosis [20].

#### 3.2.6. Role of T Helper 17 Cells

Circulating T helper 17 (Th17) cells were analyzed in three studies. One study found higher levels of circulating Th17 cells in HBV-associated HCC compared with CHB [62]; however, a second study found no difference among the two groups [28]. Additionally, a third study found the Th17/Treg ratio to be an independent risk factor of HCC development [52], and that levels of Th17 were associated with degree of liver damage. An interesting finding was that levels of PD-1 increased in patients who progressed from CHB to HCC [62].

Infiltrating Th17 cells were investigated in two studies. The first study showed lower levels of infiltrating Th17 cells in HBV-associated HCC compared with a healthy comparisons group and CHB patients [28]. The second study found that infiltrating Th17 secreted increasing levels of IL-17 along with severity of liver disease and that IL-17 promoted liver fibrosis and tumorigenesis [36].

#### 3.2.7. Role of Follicular Helper T Cells

Studies have found that levels of follicular helper T cells in HBV-associated HCC were similar compared with the CHB group and healthy comparison group [44,55,57]. However, higher expression levels of PD-1 have been shown [55] along with decreased levels of IL-21 secretion, suggesting poorer viability [57].

## 4. Discussion

This systematic review revealed that HBV-associated HCC is marked by an immunosuppressive tumor microenvironment, characterized by dysfunctional and exhausted T-cell populations. Circulating CD8+ T-cell levels were elevated in HBV-associated HCC patients compared with non-HBV HCC and healthy comparisons [30,41,54], mainly composed of terminally differentiated effector and memory CD8+ T cells [21,34]. Conversely, infiltrating CD8+ T-cell levels in the tumor microenvironment were lower in patients with HBV-associated HCC compared with liver tissue from non-HBV HCC patients and healthy comparisons [40,47,69,71]. Tumor-infiltrating CD8+ T cells in HBV-associated HCC were highly activated but dysfunctional, showing impaired cytotoxicity and exhaustion, with significantly increased expression of immune checkpoint molecules, such as PD-1, CTLA-4, LAG-3, and TIGIT. On the contrary, tumor-infiltrating HBV-specific CD4+ T cells were significantly elevated compared with all control groups (non-HBV HCC, CHB, and HC) [40,53,68,71], with Tregs being the predominant CD4+ T-cell subset [25,71]. The CD4+ T-cell populations, including Tregs, also exhibited high expression levels of immune checkpoint molecules, predominantly PD-1 and CTLA-4. Additionally, Tregs displayed high expression of the transcription factor FOXP3 [25], which plays a suppressive role in the immune system [72]. Interestingly, levels of circulating and infiltrating HBV-specific CD8+ T cells were largely similar in HBV-associated HCC compared with CHB, whereas infiltrating CD4+ T cells and Tregs were significantly elevated in HBV-associated HCC, indicating that CD4+ T-cell populations may be a dominant factor in the progression of CHB to HBV-associated HCC. A limited number of studies have investigated the role of T helper 17 cells and follicular helper T cells and found that T helper 17 cells may be associated with poor prognosis. However, no conclusions should be made based on these findings and further studies are needed to elucidate the role of these cell subsets.

The cytotoxic effects of CD8+ T cells are crucial in controlling viral HBV infection and cancer [16]. Decreasing levels of CD8+ T cells in CHB, leading towards dysfunction and exhaustion, are fundamental mechanisms in disease progression. Hepatitis B surface antigen (HBsAg) and hepatitis B core-related antigen (HBcrAg) are considered crucial factors in HBV specific immune responses and thought to be a hallmark in T-cell dysfunction. However, recent evidence has shown that the phenotypical and functional profiles of CD8+ T cells were unaffected by HBsAg levels [73]. Conversely lower HBcrAg levels correlated with higher HBV-specific CD4+ T-cell responses, indicating that HBcrAg may be a more significant viral biomarker [73]. These findings are important for the development of novel immune-based therapies. The increased enrichment of CD4+ T cells and their differentiation into Tregs have dual effects. Firstly, CD4+ T cells activate CD8+ T cells to a lesser extent. Secondly, Tregs inhibit the proliferation of CD8+ T cells. Furthermore, the expression of immune checkpoint molecules contributes to immunosuppression in the tumor microenvironment.

This systematic review highlights PD-1 expression’s crucial role in the development of HCC. Elevated PD-1 expression in circulating CD8+ and CD4+ T cells strongly correlates with disease progression and higher recurrence rates in HBV-HCC patients. PD-1 levels are notably high on exhausted CD8+ T cells in the tumor microenvironment, indicating immune dysfunction and tumor evasion mechanisms (Figure 2). Moreover, PD-1 expression correlates positively with HBV DNA levels, ALT, and AST, which is an interesting finding as HBV DNA integration in infected hepatocytes is a major driver in HCC development [74]. A study demonstrated that the blocking of TIM-3 and PD-1 restored CD8+ T-cell effector functions [42], suggesting their potential as therapeutic targets for immune checkpoint blockade strategies. The PD-1/PD-L1 axis leads to negative feedback of the immune response by blocking the T-cell receptor. Tumor cells expresses PD-L1 to avoid immunosurveillance. Immune escape is a fundamental Hallmark of cancer [75,76] and the development of PD-1 inhibitors has gained a fundamental role in the treatment of several cancers. However, the response rate to the checkpoint inhibitor nivolumab (PD-1 monoclonal antibody) for HCC is 15–20% [77], and a randomized controlled multicenter trial found no difference in treatment with nivolumab compared with sorafenib (recommended first-line chemotherapeutic drug for HCC) [78]. Monotherapy with PD-1 inhibition has thus far demonstrated questionable efficacy [79], and should be used in a personalized approach. Studies have demonstrated a higher response rate of PD-L1 inhibition in patients with low levels of HBsAg and HBcrAg [73].

The tumor microenvironment in HCC is multifactorial and complex involving HBV DNA integration, chronic inflammation, and a dysfunctional adaptive immune response, and a multi-target treatment strategy may be a potential approach in the future. Efficient preventive and curative treatment for HBV-associated HCC is lacking, and the mechanisms driving the transition towards exhausted T lymphocytes and carcinogenesis is not fully elucidated. Another important aspect is the interplay between the innate and adaptive immune responses, which is also an important driver in the persistent inflammation in CHB. Tumor-promoting inflammation is a fundamental hallmark of cancer [75,76,80,81] and it may be of great importance to characterize the inflammatory phenotypes in CHB and HBV-associated HCC to fully understand the mechanisms, and in a future perspective, identify predictive biomarkers and ultimately develop efficient treatment strategies.

### Strengths and Limitations

This systematic review included studies with general similarity regarding eligibility criteria for included study participants. Conditions other than HBV and CHB that may affect the liver and/or immune system led to exclusion across all studies, which is a strength in this review. Additionally, this review included a relatively high number of studies which collectively included 6931 participants and translational analysis was performed on biological samples with similar preparation, including PBMCs, FFPE, or fresh–frozen liver samples. Although all studies used validated platforms and demonstrated consistent results overall, a limitation of this review is the variety of methods employed to analyze T-cell activity. Differences in proteomic panels, gene expression platforms, and proliferation assays could contribute to certain discrepancies observed.

## 5. Conclusions

This systematic review highlights the immunosuppressive tumor microenvironment in HBV-associated HCC, characterized by dysfunctional and exhausted HBV-specific T cells and increased infiltration of HBV-specific CD4+ T cells, particularly Tregs. The elevated expression of checkpoint inhibitors, notably PD-1, correlates with disease progression and recurrence, suggesting its potential as a prognostic marker and therapeutic target. However, monotherapy with PD-1 inhibitors has demonstrated limited efficacy. Moving forward, a characterization of the complex interplay between the innate and adaptive immune responses holds promise for the identification of predictive biomarkers and development of new treatment strategies in HBV-associated HCC.

## Figures and Tables

**Figure 1 viruses-16-00707-f001:**
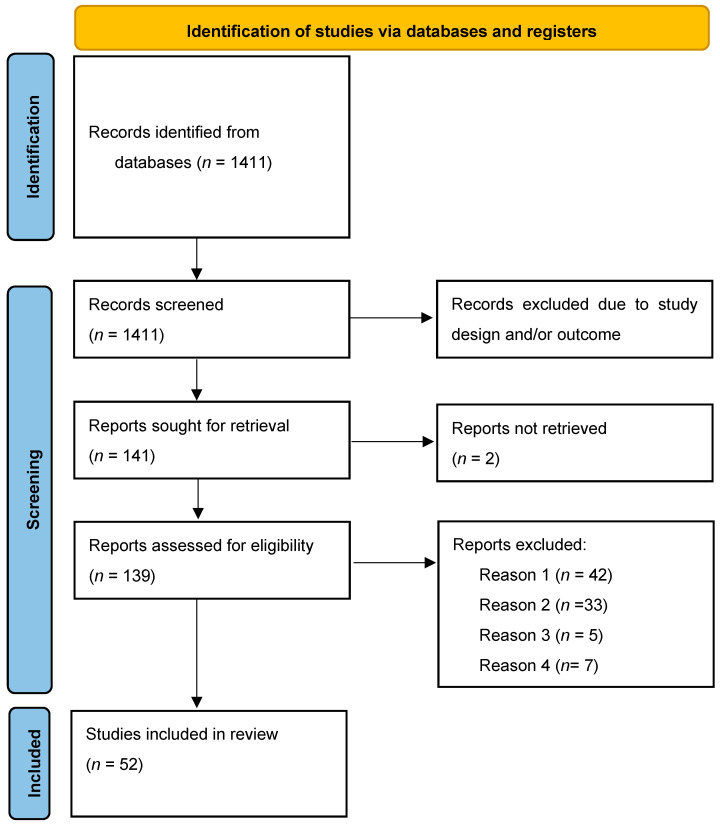
PRISMA 2020 flow diagram of the study selection process. Reason 1: studies without inclusion of human individuals ≥ 18 years of age; Reason 2: not eligible outcome; Reason 3: study without original data; Reason 4: not reported in English.

**Figure 2 viruses-16-00707-f002:**
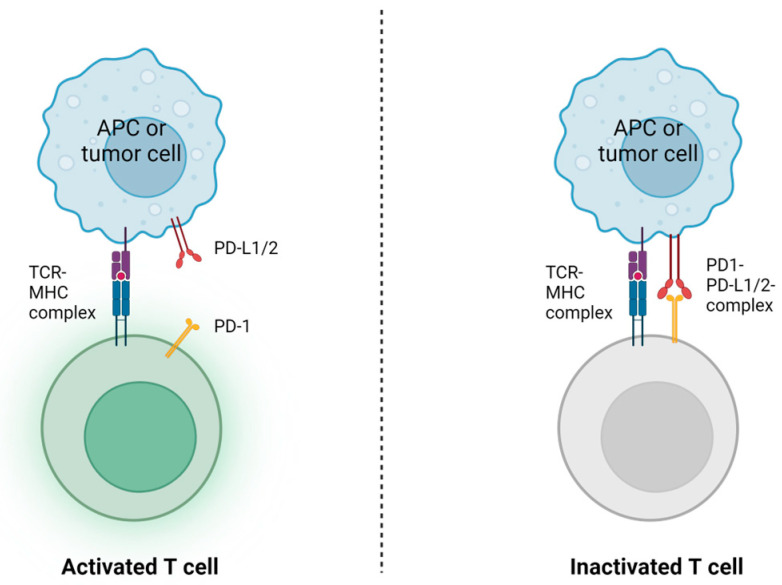
Illustration of the role of PD-1 in T-cell inactivation. PD-1 is a receptor found in T cells, which regulates immune responses by binding to its ligands, PD-L1 or PD-L2, on other cells. This interaction triggers inhibitory signals that dampen T-cell activation and effector function. This mechanism helps maintain immune tolerance but can also be exploited by pathogens and cancer cells to evade immune surveillance. Blocking the PD-1/PD-L1 or PD-L2 interaction with immune checkpoint inhibitors enhances anti-tumor immune responses, making it a potential target in cancer immunotherapy.

**Table 1 viruses-16-00707-t001:** Overview of 52 studies included in the review.

Author	Publication Year	Patients, Total (*n*)	HBV HCC	CHB	Non-HBV HCC	HC	Tissue Samples	Blood Samples
You et al. [20]	2023	58	43	0	11	4	Yes	Yes
Sun et al. [21]	2023	106	26	31	0	49	No	Yes
Liu et al. [22]	2023	25	16	0	9	0	Yes	No
Chang et al. [23]	2023	147	147	0	0	0	No	Yes
Li et al. [24]	2023	34	6	0	22	6	Yes	No
Gao et al. [25]	2022	14	7	0	7	0	Yes	No
Lu et al. [26]	2022	142	87	0	55	0	Yes	No
Li et al. [27]	2022	293	293	0	0	0	Yes	Yes
Zhang et al. [28]	2022	86	42	27	0	17	Yes	Yes
Ho et al. [29]	2021	8	29	22	0	0	Yes	No
Cheng et al. [30]	2021	46	30	0	16	0	Yes	Yes
Xin et al. [31]	2021	220	220	0	0	0	Yes	No
Zhang et al. [32]	2021	60	22	27	0	11	Yes	Yes
Liu et al. [33]	2021	152	31	78	0	43	No	Yes
Zhao et al. [34]	2020	38	19	0	0	19	Yes	Yes
Li et al. [35]	2020	60	30	0	30	0	Yes	No
Zhang et al. [36]	2020	92	49	21	0	22	Yes	No
Lim et al. [37]	2019	241	135	0	106	0	Yes	Yes
Liu et al. [38]	2019	204	122	47	0	35	No	Yes
Wang et al. [39]	2019	9	3	3	0	3	Yes	Yes
Hsiao et al. [40]	2019	1328	313	0	1015	0	Yes	No
Shen et al. [41]	2018	79	24	34	0	21	No	Yes
Liu et al. [42]	2018	90	90	0	0	0	Yes	Yes
Ou et al. [43]	2018	88	30	58	0	0	Yes	Yes
Wu et al. [44]	2018	85	18	47	0	20	Yes	Yes
Li et al. [45]	2018	8	7	0	1	0	Yes	No
Meng er al. [46]	2017	11	11	0	0	0	Yes	Yes
Huang et al. [47]	2017	411	362	0	49	0	Yes	No
Lan et al. [48]	2017	93	51	0	0	42	No	Yes
Jiang et al. [49]	2017	42	14	14	0	14	No	Yes
Liu et al. [50]	2017	160	73	87	0	0	No	Yes
Li et al. [51]	2017	32	32	0	0	0	Yes	Yes
Li et al. [52]	2017	296 *	0	0	0	0	No	Yes
Xue et al. [53]	2016	28	15	0	0	13	Yes	Yes
Liu et al. [54]	2016	815	574	0	141	100	No	Yes
Zhou et al. [55]	2016	44	20	12	0	12	Yes	Yes
Jia et al. [56]	2015	85	85	0	0	0	Yes	Yes
Duan et al. [57]	2015	33	21	0	0	11	No	Yes
Liu et al. [58]	2015	60	15	0	15	30	Yes	Yes
Dinney et al. [59]	2015	45	15	30	0	0	No	Yes
Sharma et al. [60]	2015	49	17	10	22	0	Yes	Yes
Xu et al. [61]	2014	88	16	52	0	20	No	Yes
Chen et al. [62]	2014	94	30	64	0	0	No	Yes
Li et al. [63]	2013	89	60	0	0	29	Yes	Yes
Li et al. [64]	2012	150	99	0	51	0	Yes	No
Shi et al. [65]	2011	102	56	20	0	26	Yes	Yes
Zhang et al. [66]	2010	89	49	1,5	0	25	Yes	Yes
Gehring et al. [67]	2009	30	10	20	0	0	No	Yes
Gao et al. [68]	2009	50	40	0	0	10	Yes	No
Fu et al. [69]	2007	191	123	21	0	47	Yes	Yes
Ormandy et al. [70]	2005	105	17	0	67	21	No	Yes
Total (*n*)		6931	3660	740	1627	650	36	38

HBV HCC = hepatitis B virus-associated hepatocellular carcinoma; CBH = chronic hepatitis B; non-HBV HCC = hepatocellular carcinoma with other/additional etiologies than hepatitis B virus; HC = healthy comparison group without CHB. * No patients were diagnosed with HCC at inclusion; however, patients were followed to evaluate risk of HCC development.

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
