# Peer review of "The Adaptive Immune Response in Hepatitis B Virus-Associated Hepatocellular Carcinoma Is Characterized by Dysfunctional and Exhausted HBV-Specific T Cells"

_viruses, 2024, doi:10.3390/v16050707_

Round 1

Reviewer 1 Report

Comments and Suggestions for Authors

General comment

The role of dysfunctional and exhausted cytotoxic lymphocytes in the development and persistence of carcinomas is a hot topic since many years and has gained importance with the new therapies blocking check point inhibitors of cytotoxic T cells. The systematic review summarizes data from 52 studies on the presence of T cells and the expression of markers for their dysfunction and exhaustion in patients with HBV-related HCC.

The authors divided the evaluated 6931 participants/patients into four groups: 1. chronic(?) HBV infection (CHB) with HCC, 2. CHB without HCC, 3. Non-HBV with HCC and a) HCV infection or b) nonviral causes and 4.  Healthy controls.

The question arises whether the two latter groups are useful. HCV has completely different mechanisms of immune escape from HBV and its genome does not integrate into the host genome. HCC patients without chronic virus infection have probably no foreign antigens in the liver and no obvious targets for activated T cells. The same applies for healthy controls (but see point 4b).

The presumed importance of T cell immunity in HCC implies that growth or survival of HCC cells depends on the presence of T cell antigens, may they be of viral or cellular origin. Unfortunately, the authors do not mention any of such known or presumed target antigens (point 4a).

The title claims that the evaluated studies analyzed HBV-specific lymphocytes. But the text does not mention any HBV antigen; only HBV DNA is briefly mentioned. Were the analyzed T cells stimulated with HBV antigens?

B cells and antibodies are completely neglected although they are certainly an essential part of the adaptive immune response.

Further points (Major are 2, 4, 7, 8)

1.      Spell out HBV in the title.

2.      L181, 184. Unexplained abbreviations: TIM-3, LAG-3

3.      L196. This sentence is a bit contradictory in itself. Low T-cell cytotoxicity allows for stronger HBV replication and gene expression, but reduces immune pathogenesis which is the main driver for elevated ALT, AST levels.

4.      L199. “The infiltration of CD8+ T cells was markedly decreased in tumor tissue of HBV-associated HCC compared with healthy comparisons or paired non-tumor tissue [21,41,48].” This sentence points to two problems which are not discussed in the review:

a.      HBV-infected non-tumorous hepatocytes usually express all relevant HBV antigens depending on the activity or inactivity of the antiviral T-cell response. In contrast, tumor cells are usually not competent to express HBV antigens or to replicate HBV and are HBsAg or HBcAg negative. Thus, the question arises why should CD8+ cells be present or activated in the tumor cells?

b.      Is it ethical to obtain liver tissue from healthy (!) comparisons? Do healthy livers contain large numbers of CD8+ cells?

5.      L214. Marked not market.

6.      L338. Remove one “TIGIT”.

7.      Fig. 2 is in the current form not very instructive. The text is sufficient to understand that the interaction of PD1 and its ligand leads to inactivation. The figure could be more useful if other factors mentioned in the text would be added.

8.      The main findings of the valid studies should be listed in a table. The text is difficult to grasp without further organization than subtitles.

9.      Discussion.

a.      The role of Th17 cells is not discussed although studies on this topic were included.

b.      The role of HBV gene expression is not discussed at all.  There are many reports or reviews that HBV proteins may impair activation of innate immunity, e.g. of IFN and of pathways of antigen presentation. HBeAg is known impair cellular immunity against HBV core antigen. But HBV containing hepatic tumor cells usually do not express HBeAg.

c.      The major mechanism leading to HCC is not “dysfunction and exhaustion of CD8+ cells”. It is the integration of HBV DNA which leads to uncontrolled cell growth.

Comments on the Quality of English Language

Overall, the Englishis ok with some minor errors.

Author Response

Dear Reviewer,

Thank you sincerely for the thorough review of our manuscript and for providing valuable comments. We are confident that incorporating your insights will strengthen our work. 

Based on the general comment regarding the use of control groups (HCV or non-viral HCC and Healthy comparison), we find these groups to be essential and valuable for several reasons. Firstly, all the studies included in our systematic review have utilized these control groups for comparison, which we also have done to ensure consistency and comparability. Secondly, the inclusion of control groups allows us to discern and highlight the distinctive features of the T cell response specifically within HBV-related HCC. 

Tank you for pointing out the misleading title regarding our investigation of lymphocytes. We apologize for any confusion caused and have since corrected this.

Further points 

1) Corrected 

2) Corrected 

4) a: This is indeed a valid point. The included studies in our analysis reported the levels of CD8+ T cells in tumor vs. non-tumor tissue. The role of CD8+ T cells may play a critical role in the transition from chronic HBV to HCC. Moreover, the findings from Jerome Galon's work on Immunescore highlight that the density of CD3 and CD8 in tumor tissue serve as a mroe accurate prognositic marker compared with the traditional TNM classification system. 

Given these insights, it is indeed compelling to evaluate CD8+ T cell populations in both non-tumor and tumor tissues. This dual assessment can provide valuable insights into the immune landscape associated with HBV-related HCC. 

b: All healthy controls have given informed consent before entering the protocols. Percutaneous liver biopsies are associated with limited complications. 

5: corrected 

6: removed

7: We have incorporated additional information on the function of PD-1. 

8: I agree on this point, however, it would be very challanging to summarize the results in simple tables. Due to the varied methods used across studies and the absence of reported absolute numbers, it would require seperate tables for each subtitle. 

9: a: The discussion have been revised with inclusion of Th17 

b: You make a valid point about the extensive nature of gene expression as a topic, which indeed merits a dedicated systematic review on its own. Given the scope of our current review focusing on T cells in the context of HBV-related HCC, the inclusion of gene expression analyses may not be suitable or feasible within the confines of this study. 

c: This has been included in the discussion 

Reviewer 2 Report

Comments and Suggestions for Authors

The authors performed a systematic review on the adaptative immune response in HBV-associated hepatocellular carcinoma. They found this condition associated with a dysfunction and exhaustion of HBV-specific T cells alongside an increased infiltration of HBV-specific CD4+ T cells, particularly regulatory T cells (Tregs). The information is important and well presented. Some minor details are suggested to improve the manuscript.

1.  Introduction. First paragraph: WHO mention now 296 million HBV-infected persons worldwide.

2.  I found Table 1 very useful but what was the rationale for the ordering?

3.  The last sentence of the strengths ad limitations should be rephrased and perhaps split into two sentences.

4.  The discussion of the failure of anti-PD1 treatment should benefit of inclusion of the following paper, for example: Alessio A et al. PD-1 Blockade for Hepatocellular Carcinoma: Current Research and Future Prospects. J Hepatocell Carcinoma. 2021; 8: 887–897.

Author Response

Dear Reviewer,

Thank you sincerely for the thorough review of our manuscript and for providing valuable comments. We are confident that incorporating your insights will strengthen our work. 

1) Thank you for bringing this to our attention. It has been updated

2) The order of the studies have been corrected according to year of publication

3) This has been revised. 

4) The reference has been included. 

Round 2

Reviewer 1 Report

Comments and Suggestions for Authors

The authors responded adequately to my comments and covered many (though not all)  of my suggestions in the revised manuscript.